# Diagnose-Specific Antibiotic Prescribing Patterns at Otorhinolaryngology Inpatient Departments of Two Private Sector Healthcare Facilities in Central India: A Five-Year Observational Study

**DOI:** 10.3390/ijerph16214074

**Published:** 2019-10-23

**Authors:** Elisabeth Silfwerbrand, Sumeer Verma, Cora Sjökvist, Cecilia Stålsby Lundborg, Megha Sharma

**Affiliations:** 1Department of Public Health Sciences, Global Health-Health Systems and Policy (HSP): Medicines, focusing antibiotics, Karolinska Institutet, 171 77 Stockholm, Sweden; 2Department of Otorhinolaryngology, Ruxmaniben Deepchand Gardi Medical College, Ujjain 456006, India; 3Department of Pharmacology, Ruxmaniben Deepchand Gardi Medical College, Ujjain 456006, India

**Keywords:** Otorhinolaryngology, single-dose surgical prophylaxis, chronic suppurative otitis media, diagnose-specific antibiotic prescribing patterns, private healthcare sector, India

## Abstract

Antibiotics are over-prescribed in low-and-middle-income countries, where the infection rate is high. The global paucity of standard treatment guidelines and reliable diagnose-specific prescription data from high-infection risk departments such as the otorhinolaryngology (ENT: ears, nose and throat) is a barrier to rationalize antibiotic use and combat antibiotic resistance. The study was conducted to present diagnose-specific antibiotic prescribing patterns of five years at ENT inpatient departments of two private-sector Indian hospitals. Data of all consecutive inpatients (*n* = 3527) were collected but analyzed for the inpatients aged >15 years (*n* = 2909) using the World Health Organization’s methodologies. Patient records were divided into four diagnoses groups: surgical, non-surgical, chronic suppurative otitis media (CSOM), and others. Of 2909 inpatients, 51% had surgical diagnoses. An average of 83% of patients in the clean surgery group and more than 75% in the viral and non-infectious groups were prescribed antibiotics. CSOM was the most common diagnosis (31%), where 90% of inpatients were prescribed antibiotics. Overall, third-generation cephalosporins and fluoroquinolones were most commonly prescribed. This study highlights the inappropriate prescribing of antibiotics to patients of clean surgeries, viral infections, and non-infectious groups. The single-prophylactic dose of antibiotic for clean-contaminated surgeries was replaced by the prolonged empirical prescribing. The use of microbiology investigations was insignificant.

## 1. Introduction

Antibiotics are overprescribed worldwide, but the practice of overprescribing is more common in low-and middle-income countries (LMICs), where standard treatment guidelines are not available or implemented, and infection rates are high [1,2]. Antibiotic resistance is an inevitable consequence of antibiotic consumption that limits treatment options, and increases morbidity, treatment failures, and mortality [1,2]. The World Health Organization (WHO) has recognized antibiotic resistance as a global public health threat and identified local prescribing surveillance studies as a crucial step to slow down the emergence of resistance [1,2]. National surveillance networks for prescriptions, including antibiotics, do not exist in most LMICs and India mainly due to financial, technical, and human resource constraints, and scarcity of a computerized medical record system [3]. In LMICs, where both the infection risk and use of antibiotics are high [1,2], data collection can be initiated at smaller scales. These small-scale studies are useful to inform decisions made, and over time they can be scaled up to national surveillance. Based on an estimation of national pharmaceutical sales data of 71 countries in 2010, India was the largest antibiotic consumer for humans [4]. Thereby, India is also a presumed hub of antibiotic-resistant bacteria.

Antibiotics are prescribed profusely at healthcare facilities. These places also provide suitable conditions for rapid emergence and spread of antibiotic-resistant bacteria [3,5,6]. The few available surveillance studies from India primarily focus on public sector healthcare facilities [5,7,8,9,10]. However, a majority of the Indian population seek healthcare at private sector facilities, where the poor implementation of globally available guidelines and lack of local prescribing guidelines is evident [5,8,11,12]. Studies from high-income countries report the high prescribing of antibiotics to patients admitted to otorhinolaryngology departments, commonly known as the department of ear, nose, and throat or otorhinolaryngology (ENT) [13,14]. In India, the patterns of antibiotic prescribing for ENT diagnoses have only been published from outpatient departments and primary healthcare facilities [15,16,17] and not from inpatient departments of higher healthcare levels. 

### Aim 

The aim of this study is to present and compare antibiotic prescribing patterns corresponding to ENT diagnoses (indications), and further probing of surgical and non-surgical indications, at the ENT departments of two private sector, tertiary healthcare hospitals: a teaching hospital (TH) and a non-teaching hospital (NTH), in the Ujjain district of Central India.

## 2. Materials and Methods 

### 2.1. Study Design

A prospective study was conducted for five years (2008–2013), including all patients admitted in the ENT inpatient departments at the TH and the NTH. The data were analyzed and presented at the department level, diagnosis groups, and specific diagnosis level for the two hospitals.

### 2.2. Study Settings

The two study hospitals are located in the Ujjain district of Madhya Pradesh state in Central India. The study settings have been described in previous publications in detail [5]. In brief, the study settings were a TH that is associated with a private medical college and an NTH. The TH is located in a rural area of Ujjain district and had 570 hospital beds at the time of the study. The healthcare policies of the TH were comparable to the Indian public-sector healthcare facilities as it provided medical care and medicines on a full charity basis to all patients visiting the hospitals. The NTH is situated in the city area of Ujjain (350 beds) where patients paid the subsidized cost for medical consultations, prescribed medicines, and diagnostics. Diagnostic facilities, including microbiology laboratories, were readily available at both study settings. Diagnose-specific standard treatment guidelines were not available at any of the study hospitals at the time of data collection.

### 2.3. Data Collection and Process of Analysis 

The data collection has been explained in detail in previous publications [5,8]. In short, the data collection tool was developed specifically for the study. Basic demographics, admission and discharge dates, description of prescribed antibiotics such as name, dose, duration, and frequency, and the diagnoses were recorded manually by nurses. The nurses were trained for manual data collection using the contextualized data collection tool. The study population consisted of all consecutive patients admitted to the ENT departments of the two study hospitals. The data were entered in Epidata software 3.1 (The EpiData Association, att. Jens Lauritsen, Enghavevej 34, DK5230 Odense M, Denmark, Europe)) (and Excel sheets.

Prescribed antibiotics were classified according to the WHO recommended methodologies (i.e., the Anatomical Therapeutic Chemical classification system (ATC) and the defined daily dose (DDD)) to facilitate international comparison of the results [18]. The DDD/100 prescriptions was used as a unit to present DDD prescribed, where one prescription denotes the amount of an antibiotic substance prescribed to a patient in one day.

The indications assigned by the treating consultants in the patients’ files were recorded at the time of discharge and were not verified externally. The indications were classified according to the International Classification of Diseases version 10 (ICD-10) [19]. 

The study was focused on the anatomical and therapeutic main groups of the antibacterials for systemic use, with the WHO suggested ATC code-J01 [18]. The rate of diagnose-specific antibiotic prescription per patient and the percentage of antibiotic prescriptions adherent to the WHO Model Lists of Essential Medicines (WHOLEM) [20] and the National List of Essential Medicines of India (NLEM) were calculated [21]. The oral metronidazole was classified with the systemic metronidazole with the ATC code of J01XD01, as indicated in NLEM [21]. For the non-listed new fixed-dose combinations (FDC) of antibiotics, ATC codes were used as assigned by Sharma et al., using J01RA [5]. Descriptive analysis was performed using Microsoft Excel and SPSS Version 23 (IBM Corp., SPSS Statistics for Windows, Armonk, NY, USA). Frequencies and percentages were calculated for categorical variables and the sum, median, and range for numerical variables. Pearson’s chi-squared test was used to test the significance when comparing categorical variables. The level of significance was set to *p* < 0.05.

Data of (i) the inpatients aged less than 15 years old, (ii) those that did not have an ENT-related diagnosis, or (iii) had incomplete patient records were excluded before analysis (*n* = 550, Figure 1). Patients who stayed for at least one night in the ENT wards were considered as inpatients and were included in the analyses. The inpatients were grouped, based on the assigned diagnoses, according to the guidelines of the United States of America (USA) and the Scottish Intercollegiate Guidelines Network (SIGN) to facilitate global comparison [22,23]. Four diagnoses groups were formed, namely, group A, B, C, and D (Figure 1, Appendix A). Group A comprised all inpatients with an indication of surgery. Group A was further divided into three subgroups based on the ENT surgical wounds classification system [22,23] (Table 1 and Figure 1): subgroup A1—surgery in an area of manifest infection (i.e., contaminated/dirty surgery); A2—surgery in non-sterile tissue (i.e., clean-contaminated surgery); and A3—clean surgery. Group B contained all non-surgical diagnoses, which were further divided into three subgroups: B1—clinical infection of bacterial, fungal, or parasitic origin; B2—clinical infection of viral origin; and B3—non-infectious indications, as presented in Figure 1. Antibiotic treatment was recommended for subgroups A1 and B1 and thus was not analyzed extensively. In the present study, we used the definition of surgical prophylactic antibiotic treatment, as presented in the SIGN classification system (i.e., the use of antibiotics before, during, or after a diagnostic, therapeutic, or surgical procedure to prevent infectious complications [22]).

The prevalence of chronic suppurative otitis media (CSOM) is high in the West Pacific, South East Asia, and particularly in India [24]. Antibiotics are commonly prescribed to patients with CSOM [24]. Therefore, prescriptions to CSOM patients might influence the overall results of surgery patients. Thus, all patients with CSOM were categorized separately in group C. Group D contained other indications that could not be categorized in groups A, B, and C such as atrophic rhinitis. The prescriptions for group D were not analyzed in detail.

### 2.4. Ethics Approval

The present study was observational, and all measures were in place to maintain patients’ confidentiality during data collection as well as analysis in accordance with the standard guidelines. None of the patients was contacted at any stage of the study, and no patient was identified or analyzed individually. The data were analyzed collectively forming various groups. The ethics committee of R. D. Gardi Medical College, Surasa, Ujjain, India approved the study with approval number: 41/2007 and 114/2010. Following the norms of an observational study design, the need for the approval of trial registration was waived by the ethics committee. 

### 2.5. Availability of Data and Material

The datasets generated during the current study are not publicly available due to breaching individual privacy, but de-identified datasets are available from the corresponding author on reasonable request. A summary of data analyzed during this study is included in this published article and its Appendix A.

## 3. Results

Of the 3527 patients admitted to the ENT departments, 2909 were included in the analysis (Figure 1). Out of included inpatients, 2358 underwent surgical procedures. Of 2358 inpatients, 883 had CSOM (group C, Table 2) and the rest were categorized in group A. Group B, the non-surgical indications, consisted of 537 inpatients whereas 14 inpatients had other diagnoses (group D). In both hospitals, CSOM was most common among patients aged between 15 and 50 years, and cancers were most common among the patients aged more than 50 years.

Overall in group A 85% were prescribed antibiotics. At diagnosis group level, antibiotic prescribing was higher specifically in the dirty/contaminated surgery (A1) and clean-contaminated surgery (A2) subgroups at the NTH than at the TH (96% each compared to 94% and 93%), while it was higher in the clean surgery (A3) at the TH than at the NTH (86% compared to 81%, Table 2). The DDD/100 prescriptions were higher at the NTH for all diagnoses groups compared to the TH (*p* < 0.05). Overall, the prescriptions made using generic names were significantly higher (*p* < 0.05) and adherence to the WHOLEM and NLEM were also higher at the TH than at the NTH (Table 2).

A diagnose-specific analysis of tonsillectomy and adenoidectomy in subgroup A3 showed that at the NTH, 36 patients underwent any of these surgeries and 32 were prescribed antibiotics with a median treatment period of three days (range: 1–6 days). At the TH out of the 109 patients who underwent tonsillectomy or adenoidectomy, 106 patients were prescribed antibiotics with a median treatment period of six days (range: 1–21 days). Overall, at the NTH, the proportion of prescriptions made using brand names (94%, Table 2) was higher when compared to that at the TH (55%). A higher proportion of the prescriptions of subgroups A2 and A3 at the NTH adhered to the international prescribing guidelines (20–25%, Figure 2) than at the TH (10–11%) [22,23]. 

CSOM (group C) was the most common diagnosis with 20% of total inpatients at the NTH and 33% at the TH (Table 2). At both hospitals, group C had the most extended duration of hospital stay and antibiotic treatment. Both durations were longer at the TH with a median of 11 days (range: 1–38 days) and four days (range: 1–32 days) at the NTH. At the TH, 95% of the inpatients with CSOM were prescribed antibiotics and 89% at the NTH. One patient with CSOM at the NTH was prescribed antibiotic ear drops.

In the non-surgery group (group B), antibiotics were prescribed to a higher proportion in all subgroups at the TH (83–94%) than at the NTH (75–92%, Table 2). In the non-infectious group (subgroup B3), 75% were prescribed antibiotics at the NTH and 83% at the TH.

Figure 3 presents 90% of the most commonly prescribed antibiotic substances (5th level of ATC) at the diagnosis subgroup level, based on the drug utilization 90% method (DU90%) [18,25]. At the department level, ceftriaxone and FDCs were the most prescribed antibiotics at the NTH and ceftriaxone and ciprofloxacin at the TH for the selected diagnoses groups. Of the total prescriptions, 24% at the NTH and 18% at the TH were FDCs. A second-generation cephalosporin, cefuroxime (J01DC02), was prescribed at the NTH in subgroup A2 (17%), but not at all at the TH. Among CSOM patients, ceftriaxone was most commonly prescribed in both hospitals, followed by ciprofloxacin in the TH and FDCs in the NTH. 

The duration of antibiotic prescription was highest in groups A and C. Figure 4 presents the duration of antibiotics prescription in days to the inpatients of groups A and C at the subgroup level. Among the patients in subgroup A2 (clean-contaminated surgery), 96% at the NTH and 93% at the TH were prescribed antibiotics for a longer duration than the recommended duration for this group (24 h).

At discharge, a significantly higher number of patients were prescribed antibiotics at the TH (43%) than at the NTH (33%, *p* < 0.05). The patients in the surgical diagnosis groups (A and C) were more often prescribed antibiotics than the patients in non-surgical group B. Overall, the FDCs of amoxicillin with clavulanic acid J01CR50 (45%), cefuroxime J01DC02 (19%), and ciprofloxacin J01MA02 (8%) were the most commonly prescribed antibiotics at discharge. 

Overall two deaths were reported at the TH, one with CSOM and the other with tuberculosis. No comorbidities were reported on these patients. Overall, the samples of 0.5% inpatients were sent for bacterial culture and antibiotic susceptibility tests from both settings. In total, the antibiotic therapy was changed in 0.4% of prescriptions based on the microbiology culture and susceptibility reports. 

## 4. Discussion

To the best of our knowledge, this is the first long-term study at diagnosis level that describes the pattern of antibiotic prescribing among ENT inpatients of Indian hospitals. Due to lack of ENT department-specific studies from LMICs, we compared our results with studies of other departments. The present study highlights that antibiotics were prescribed to almost nine of the ten admitted patients (TH 91% and NTH 86%). Antibiotics were prescribed most frequently and for the most prolonged period to CSOM inpatients. Antibiotics were commonly prescribed for clean surgeries, clinically viral infections, and non-infectious diagnoses. Overall, empiric treatment was extended during the entire hospital stay for most inpatients. One of the reasons for this extension might be a lack of sending samples for bacterial culture and antibiotic susceptibility tests. None of the inpatients was prescribed a single prophylactic dose of antibiotics for clean-contaminated surgeries, as per recommendations [22,23]. Ceftriaxone and ciprofloxacin were predominantly prescribed in all groups. In India, the prevalence of bacteria resistant to carbapenems is increasing [4]. However, carbapenems were not prescribed in the ENT departments of the study. In all diagnoses groups, the DDD/100 prescriptions were higher at the NTH than at the TH.

### 4.1. Prescribing Patterns in Surgical Diagnoses Group, Group A

Among the patients who had clean-contaminated surgery, subgroup A2, a single dose regime of a first- or second-generation cephalosporin, preferably cefazolin, is recommended as prophylactic antibiotic treatment [22,23]. In our study, a third-generation cephalosporin was the preferred antibiotic choice in both settings. Similar results were shown in two studies conducted at a general surgery department in Western India [26] and at a private tertiary healthcare hospital in Southern India [27], where third-generation cephalosporins were reported as the most commonly prescribed class of antibiotics. The preference for a broader spectrum antibiotic, such as the third-generation cephalosporin, might be due to the overall increase in the prevalence of bacterial resistance to antibiotics in India that has been observed in other settings [4]. De-escalation of antibiotic therapy is suggested for settings where broad-spectrum antibiotics are the first drug of choice for surgical diagnoses but was not found at our settings [28]. Overall, nine out of ten patients who underwent clean-contaminated surgery were prescribed prophylactic antibiotic treatment for more than one day. This might be due to a common global misunderstanding, as presented by Bratzler D et al., that longer prophylactic antibiotic treatment periods are considered to be more effective to prevent surgical site infections (SSI) than single-dose regimes [23]. Prescribing broad-spectrum antibiotics during a prolonged period as prophylactic antibiotic treatment is not recommended due to the increased risk of adverse effects, the risk of emergence of antibiotic resistance, and higher treatment costs [1,6,23]. Studies to advocate prescribing a single dose of a first- or second-generation cephalosporin and to develop and introduce local prescribing guidelines based on susceptibility patterns are also suggested.

Among the patients who underwent clean surgery (subgroup A3), 81% at the NTH and 86% at the TH were prescribed antibiotics, even though prophylactic antibiotic treatment is not recommended for this subgroup [22,23]. Khan et al. from Southern India, reported that many surgeons often take a ‘safety approach’ and prescribe antibiotics to minimize the risk of SSI in clean surgeries [27]. It is worth mentioning that the risk of SSI in clean surgery is reported to be less than 2% [22]. Reasons to use a ‘safety approach’ could be a high patient burden per physician or presumed poor hygienic conditions of the patients as the catchment area included villages of low socio-economic status [28]. All these presumptions need to be verified through a suggested qualitative study.

The results of antibiotic prescribing practices for tonsillectomy and adenoidectomy in the subgroup A3 highlights several underlying issues. The first issue was in accordance with a globally ongoing discussion. The discussion raises the question of whether antibiotics should be prescribed as a prophylactic antibiotic treatment to patients undergoing tonsillectomy or adenoidectomy or not [22,23,29]. One side of the debate is presented by the USA and SIGN guidelines. These guidelines suggest the prescribed antibiotics as general prophylactic antibiotic treatments for the above mentioned surgical procedures, but only if the patient exhibits risk factors of acquiring bacterial infections [22,23]. The other side of the debate presents a regional scenario based on local infection risk factors, as mentioned in a textbook of ENT diseases written by Indian authors [29]. According to the recommendations of the book, prophylactic antibiotic treatment can be prescribed post-operatively up to a week [29]. The ENT physicians at our study settings might have followed the local suggestion of prescribing antibiotics for the surgeries in question, and this could also explain the overall high proportion of antibiotic prescriptions in subgroup A3. 

Another issue raised was regarding universal applicability of a guideline. In the present study, the American and Scottish guidelines were used to facilitate the classification of the surgical indications concerned with prescribing of antibiotics [23]. On the other hand, the local prescribing rationale suggests classifying tonsillectomy and adenoidectomy in subgroup A1 (i.e., dirty/contaminated surgery) and not in A3 (i.e., clean surgery) [29]. Moreover, the use of different surgical techniques and varied access to resources in a setting also affect the risk of surgical site infections and are causes for prescribing antibiotics [30]. Hence, the present diagnoses-specific study highlights our concern of applicability of available international guidelines, often based on surveillance data of high-income countries, to the other parts of the world, especially LMICs. The antibiotic prescribing patterns of subgroup A3 supports the WHO’s emphasis on development and implementation of local standard treatment guidelines based on local surveillance data of prescriptions, relevant risk factors for infections, infrastructure of the healthcare system, and availability of resources, to combat against antibiotic resistance [2].

### 4.2. Prescribing Patterns to Inpatients with Non-surgical Diagnoses, Group B

Among the inpatients with non-surgical diagnoses, group B, antibiotic treatment was indicated only to patients having a microbiologically confirmed or clinically estimated high risk of a bacterial infection (subgroup B1) [22,23]. A majority of the patients in subgroup B1 of both hospitals were prescribed antibiotics as per the recommendations. However, antibiotics are not indicated for infections of suspected viral origin (subgroup B2) and for non-infectious diseases (subgroup B3) [31]. Still, 76% of the patients at the NTH and 84% patients at the TH in these subgroups were prescribed antibiotics. There is a need to conduct a separate probing study to explore the underlying factors for this practice.

Furthermore, a longer duration of hospital stay has been reported to correspond to a higher risk of acquiring healthcare-associated infections (HAI) and increases the risk of being prescribed antibiotics [32]. Thus, in the present study, a preventive approach for HAI might be the reason for the higher proportion of patients being prescribed antibiotics at the TH compared to the NTH. However, higher antibiotic doses (DDD/100 prescriptions) were prescribed at the NTH compared to the TH. Prescribing antibiotics for non-indicated conditions, at higher doses and for more extended periods than indicated, are considered as preventable factors that, if continued, might accelerate the development of antibiotic resistance [1,2,6]. 

### 4.3. Prescribing Patterns to CSOM Inpatients, Group C

CSOM was the most common diagnosis in our study settings. The procedures related to CSOM are categorized in classes comparable to subgroup A1 (dirty/contaminated surgery, e.g., emergency mastoidectomy), to which antibiotics should be prescribed as treatment and A3 (clean surgery, e.g., tympanoplasty), to which antibiotics should not be prescribed [22,23,33]. In group C, antibiotics were prescribed to 95% of inpatients at the TH (NTH 89%), and 91% were prescribed antibiotics for more than 48 h (NTH 74%). The difference between the hospitals could be due to the presence of a higher number of contaminated/dirty surgeries performed at the TH than at the NTH. Furthermore, a WHO review article on CSOM, reports that topical antibiotics are superior to systemic antibiotics in terms of efficacy and have an advantage of less contribution to the development of antibiotic resistance [24]. However, only one patient was prescribed antibiotic ear drops. 

The inpatients in group C at the TH received antibiotics for the most prolonged period with a median of 11 days. The lack of comparable antibiotic surveillance studies and local antibiotics prescribing guidelines for CSOM surgeries at the ENT inpatient departments of the South-Asian region are the major factors affecting such a prescribing pattern. We recommend conducting qualitative studies to further analyze the factors driving the prolonged prescribing of antibiotics and infrequent prescribing of topical antibiotics.

The new FDCs of antibiotics were prescribed at a lower extent (J01RA, Figure 3) as compared to studies from other departments in the study settings [5,34]. Most of the new FDCs of antibiotics have no underlying scientific justification and do not add to a drug’s efficacy; however, they add to the cost of therapy, increase adverse effects, and encourage antibiotic resistance [35]. Thus, less prescribing of new FDCs in both study settings could be appreciated. 

The samples for bacterial culture and antibiotic susceptibility tests were not sent frequently in both study settings. Similar results were found in a short-term study conducted previously at the study hospitals [5]. Empiric treatment is used to start antibiotic therapy for a suspected bacterial infection after collection of a sample from the suspected infection site for susceptibility and culture tests [6]. The empirical prescribing ought to be reassessed based on the microbiological test results [6]. Therefore, underutilization of microbiology tests might have resulted in extended empirical prescribing to the inpatients throughout the hospital stay. 

Active CSOM infections are often polymicrobial, thus, most of the patients might have received several antibiotic prescriptions at the lower levels of healthcare facilities before visiting and being admitted at the tertiary level healthcare settings (i.e., in the study hospitals) [24,33]. The high risk of multiple antibiotic treatments before hospital admission might be the reason for a low threshold for sending samples before initiating an empiric antibiotic therapy at the study settings [6]. The factors such as continuous medical education (CME) sessions and prohibition against interacting with the pharmaceutical representatives at the TH might have influenced overall judicious prescribing of antibiotics in the TH compared to the NTH [5]. 

Results of our study highlight the need to conduct similar surveillance studies at other settings followed by contextualized qualitative studies to understand the underlying factors affecting practitioners for the observed patterns of antibiotic prescribing.

The strengths of this study were its long-term, continuous data collection, and detailed data at a diagnosis level recorded for all admitted patients. Furthermore, the data were from the private sector that provides healthcare to most of the Indian population. A limitation of this study was that perioperative notes were not included, which could have facilitated the interpretation of the results. However, this was not the objective of the present study. Although the data collection process was supervised robustly, the possibility of human error during data collection and entry cannot be denied. However, the possible effect of such human errors was expected to be minor due to the large study population, long study duration, robust monitoring, trained data recorders, and data entry staff. The establishment of the manual process of data collection, coding of antibiotics, indications, and data entry explains why the data was analyzed and published at the current time. The recommended denominator for DDD is per 100 bed days for the inpatients [18]. However, in the present study, the unit of data collection was a prescription therefore, the DDD/100 prescriptions was used to calculate amount of antibiotics prescribed. Moreover, this unit is more feasible to apply to similar drug utilization studies of selected diagnoses. 

## 5. Conclusions

The present study offers a unique insight into the otorhinolaryngology inpatient departments of private healthcare facilities in India and identifies several under-focused issues to be addressed in the future. In the present study, a high proportion of inpatients with non-indicated diagnoses, such as clean surgeries, viral infections, and non-infectious indications were prescribed antibiotics. A significant deviation was observed between the antibiotic prescribing patterns at the study settings and the international recommendations and guidelines. This deviation indicates poor applicability of these recommendations in LMICs that are not based on the data from LMICs but form high-income countries. This once again highlights the need for the development of contextualized standard treatment guidelines based on local infectious diseases and prescribing data. 

The scarcity of local antibiotic prescribing surveillance studies and resistance patterns limits the development of local prescribing guidelines. Surveillance studies at the private healthcare settings of LMICs are crucial. Our sustainable, low-cost solution of manual data collection could be adapted at resource-constrained settings where there is a lack of computerization to conduct surveillance studies. Furthermore, the use of ATC/DDD methodology and the ICD-10 codes made our results comparable and exchangeable to other global studies. The findings of this study formulate a standard policy and indicate the practice patterns of antibiotic prescribing in high-risk infection inpatient departments. Prescribing broad-spectrum antibiotics empirically and inadequate use of the microbiology laboratory was also observed in both study hospitals.

Further probing using a qualitative approach is suggested for better understanding the influencing factors for current prescribing patterns, under-utilization of microbiology facilities, and possible areas of improvement. We also recommend developing a customized educational workshop and group discussions as a part of antibiotic stewardship programs for prescribers focused on developing and implement ingdiagnose-specific local antibiotic prescribing guidelines.

## Figures and Tables

**Figure 1 ijerph-16-04074-f001:**
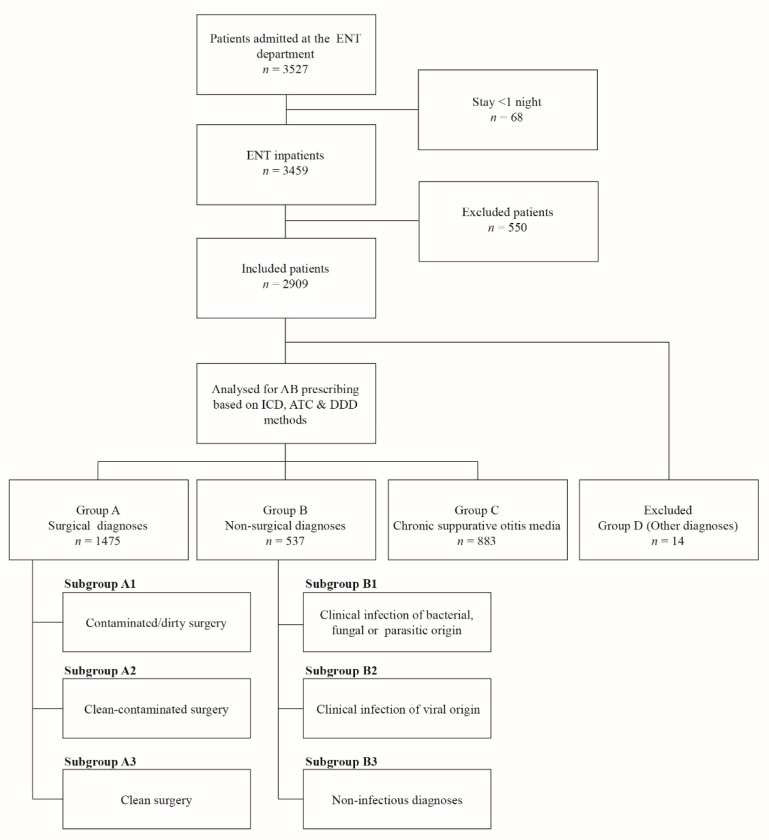
The division of diagnoses in groups (A–D) and subgroups (1–3). Abbreviations: ATC = Anatomical Therapeutic Chemical classification system; DDD = Defined daily dose; ENT = Ear, nose and throat (otorhinolaryngology); ICD-10 = International Classification of Diseases, version 10 by the World Health Organization (WHO); *n* = number of patients.

**Figure 2 ijerph-16-04074-f002:**
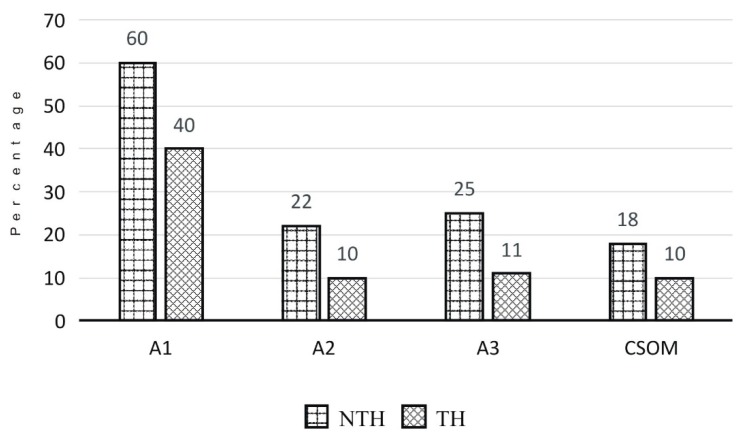
The percentage of adherence to the international antibiotic prescribing guidelines* for ENT surgeries and CSOM in the ENT departments at the study hospitals. *The guidelines for antimicrobial prophylaxis by Bratzler D [22] and the international prescribing guidelines refer to the Scottish Intercollegiate Guidelines [23]. Abbreviations: A1 = contaminated/dirty surgery; A2 = clean-contaminated surgery; A3 = clean surgery; CSOM = Chronic suppurative otitis media; ENT = ear, nose and throat (otorhinolaryngology); NTH = non-teaching hospital; TH = teaching hospital.

**Figure 3 ijerph-16-04074-f003:**
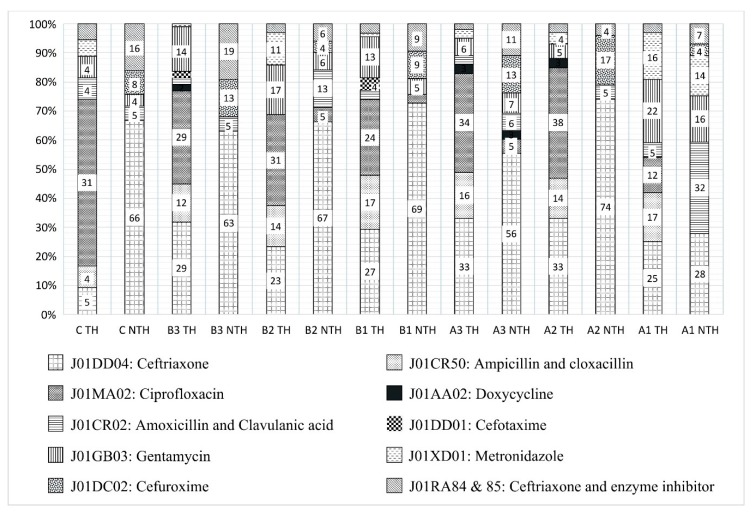
Selection of prescribed antibiotics at the diagnosis subgroup level. In the graphs, 100% represents 100% of the antibiotics prescribed based on the drug utilization 90% method (DU90) by Bergman et. al. [25]. Numeric representation is done for the antibiotics prescribed in more than 3% of prescriptions. Abbreviations: A1 = contaminated/dirty surgery; A2 = clean-contaminated surgery; A3 = clean surgery; B1 = clinical infection of bacterial, fungal or parasitic origin; B2 = clinical infection of viral origin; B3 = non-infectious diseases; C = group C (i.e., chronic suppurative otitis media); NTH = non-teaching hospital; TH = teaching hospital.

**Figure 4 ijerph-16-04074-f004:**
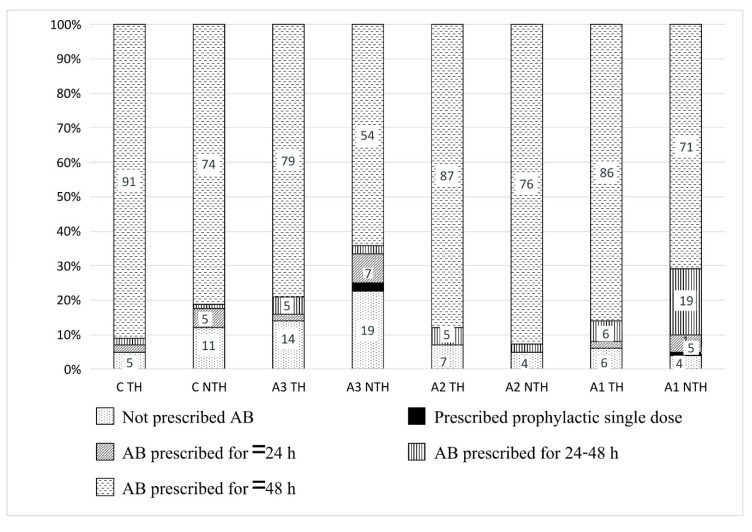
Duration of antibiotic treatment to the patients of groups A and C in the study hospitals. Numeric representation is done for the antibiotics prescribed in more than 3% of prescriptions. Abbreviations: A1 = contaminated/dirty surgery; A2 = clean-contaminated surgery; A3 = clean surgery; AB = antibiotics; C = group C (i.e., chronic suppurative otitis media); h = hours; NTH = non-teaching hospital; TH = teaching hospital.

**Table 1 ijerph-16-04074-t001:** The categories of the ENT surgical wounds and the risk of surgical site infections as per international recommendations *.

Categories	The Surgical Field at the ENT Department	Risk of Infection	Recommendation to Prescribe Antibiotics
Contaminated/dirty	Areas of manifest infections, open wound >4 hours.	>20%	Yes
Clean-contaminated	Penetration of mucosa in the oral cavity, pharynx, larynx, esophagus, or nasal cavity. Radical cancer surgery.	<10%	Yes, as prophylaxis
Clean	Sterile tissue or tissue that can be made sterile (i.e., ear, sinus or skin), tonsillectomy, and adenoidectomy.	<2%	No

* Scottish Intercollegiate Guidelines [23] Abbreviation: ENT = Ear, nose and throat (otorhinolaryngology)

**Table 2 ijerph-16-04074-t002:** Antibiotic prescription patterns at diagnosis group level at the ENT departments at the two study hospitals.

	Surgical Diagnoses (Group A)				Non-Surgical Diagnoses (Group B)			CSOM (Group C)
		Contaminate/Dirty Surgery (A1)	Clean-Contaminated Surgery (A2)	Clean Surgery (A3)	Clinical Infection of Bacterial, Fungal or Parasitic Origin (B1)	Clinical Infection of Viral Origin (B2)	Non-Infectious Diseases (B3)	C
Total inpatients, *n* (%)	NTH	101 (15)	45 (7)	198 (29)	65 (10)	37 (5)	89 (13)	140 (20)
	TH	182 (8)	270 (12)	679 (31)	96 (4)	42 (2)	208 (9)	743 (33)
Inpatients prescribed antibiotics, *n* (%)	NTH	97 (96)	43 (96)	162 (81)	60 (92)	29 (78)	67 (75)	125 (89)
	TH	171 (94)	250 (93)	584 (86)	90 (94)	37 (88)	173 (83)	706 (95)
Total antibiotic prescriptions made for inpatients, *n* (%)	NTH	509 (20)	190 (8)	657 (26)	248 (10)	98 (4)	251 (10)	569 (23)
	TH	2261 (11)	2224 (11)	5302 (26)	889 (4)	251 (1)	1727 (9)	7648 (38)
DDD/100 prescriptions	NTH	172.2	164.7	156.7	129.4	147.5	145.2	126.9
	TH	75.8	89.8	85.4	82.6	80.3	87.9	90.9
Percentage of prescriptions adherent to WHOLEM	NTH	86	75	74	76	92	66	76
	TH	82	83	80	80	84	85	90
Percentage of prescriptions adherent to NLEM	NTH	85	75	75	78	85	68	75
	TH	80	81	80	80	84	85	90
Percentage of AB prescribed using generic names	NTH	14	4	5	4	2	2	5
	TH	46	37	39	39	37	42	54
Percentage of AB prescribed via parenteral route	NTH	99	94	85	96	85	94	96
	TH	85	59	64	72	68	68	68
Duration of hospital stay in days, median (range in days)	NTH	3 (1–16)	4 (2–17)	3 (1–25)	3 (1–11)	2 (1–6)	3 (1–10)	4 (1–32)
	TH	7 (1–40)	7 (1–48)	8 (1–77)	8 (1–22)	5 (1–36)	7 (1–58)	11 (1–38)
Duration of AB treatment in days, median (range in days)	NTH	3 (1–12)	4 (2–9)	3 (1–25)	4 (1–11)	3 (1–6)	3 (1–9)	4 (1–14)
	TH	8 (1–23)	7 (1–47)	8 (1–64)	8 (1–23)	5 (1–11)	7 (1–57)	11 (1–36)

Abbreviations: AB = antibiotics; CSOM = chronic suppurative otitis media; ENT = ear, nose and throat (otorhinolaryngology); *n* = number; NLEM = National List of Essential Medicines in India; NTH = non-teaching hospital; TH = teaching hospital; WHOLEM = WHO Model List of Essential Medicines.

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
