# Peer review of "Diagnose-Specific Antibiotic Prescribing Patterns at Otorhinolaryngology Inpatient Departments of Two Private Sector Healthcare Facilities in Central India: A Five-Year Observational Study"

_ijerph, 2019, doi:10.3390/ijerph16214074_

Round 1

Reviewer 1 Report

The study was to evaluate the antimicrobial prescribing patterns in the ENT departments of two private hospitals in India. The information is important to identify the areas of improvement and to promote appropriate antimicrobial use in this area with lack of resources. I have only minor comments/edits following:  

Abstract.

Line28-29. Change “unindicated” to “inappropriate”

Introduction

Line 36. Change “the practice” to something like “this overprescribing practice.”

Lines 57-58. Remove the sentence starting with “Therefore, surveillance…. is crucial” which seems unnecessary.

Results/discussion

Table 2. Authors estimated DDD/100 prescriptions. 100 prescriptions is not a common denominator to assess drug utilization. Why did authors choose to use this? The difference of DDD/100 prescriptions in NTH setting vs. TH setting seems significant. Please discuss the interpretation in the result/discussion. Table 2 – Clarify “total prescriptions”. Is it Antibiotic total prescription? Overall, authors presented the data in NTH and TH and compared them. However, authors do not report whether the difference is significant and why the difference was possibly resulted in. I recommend authors include this discussion in the text to further identify the areas of opportunity.

Line 176. “the drugs of choice at the NTH”  Drugs of choice for what condition?

Lines 280-282 – rewirte the sentence to improve clarity – something like “However, ……viral origin (e.g. subgroup B2) or no-infectious disease (e.g. subgroup B3).”

Lines 304-308 – The sentence starting with “The lack of comparable….” Does not read well. Clarify what  the following means. “…restrict the possibilities to describe the reasons for this pattern…”

Line 314. -What does “the low practice” mean? Infrequent ordering of culture & susceptibility?

Line 316. – “in the settings” – clarify what the settings are? Clarify.

Conclusions:

Line 341-343. The sentence starting with “A deviation in diagnose…. In the LMICs” is not clear what it means. Clarify.

Author Response

The authors are thankful for the thoughtful and thorough review. We have now modified the manuscript based on your suggestions and comments. 

Please see the attachment for a point-wise response to your valuable suggestions and comments.

Reviewer 2 Report

Overall, the article was interesting and covered a very important topic that is understudied. 

The study did an excellent job of describing relevant background and previous studies done. 

Would it be possible to clearly describe the study design? assuming it was a retrospective cross-sectional study.  

Would it be possible to have a descriptive chart or figure that further described the results, the results are very hard to follow

Table 2 is very unclear and cluttered, consider creating a more descriptive pie graph that would explain the information that will be more reader friendly. 

It would be great to incorporate the patterns the study discovered with the antimicrobial usage.

Minor revisions

Line 40 "reconised" change to recognize  

Author Response

We thank the reviewer for the thoughtful review and kind words. We have
revised the manuscript based on the suggestions made. 

A point-wise response to your valuable suggestions and comments could be seen in the attached document.
